# CD163-Expressing Porcine Macrophages Support NADC30-like and NADC34-like PRRSV Infections

**DOI:** 10.3390/v14092056

**Published:** 2022-09-16

**Authors:** Yulin Xu, Mengxue Ye, Shaohua Sun, Qi Cao, Jia Luo, Yuening Wang, Wanglong Zheng, François Meurens, Nanhua Chen, Jianzhong Zhu

**Affiliations:** 1College of Veterinary Medicine, Yangzhou University, Yangzhou 225009, China; 2Joint International Research Laboratory of Agriculture and Agri-Product Safety, Yangzhou 225009, China; 3Comparative Medicine Research Institute, Yangzhou University, Yangzhou 225009, China; 4Jiangsu Co-Innovation Center for Prevention and Control of Important Animal Infectious Diseases and Zoonoses, Yangzhou University, Yangzhou 225009, China; 5BIOEPAR, INRAE, Oniris, 44307 Nantes, France; 6Department of Veterinary Microbiology and Immunology, Western College of Veterinary Medicine, University of Saskatchewan, Saskatoon, SK S7N 5E2, Canada

**Keywords:** porcine reproductive and respiratory syndrome virus, cell tropism, CD163, MARC-145 cells, stable macrophages

## Abstract

Porcine reproductive and respiratory syndrome virus (PRRSV) has a strict cell tropism. In addition to the primary alveolar macrophages, PRRSV is strictly cytotropic to African green monkey kidney cells, such as MARC-145 cells; however, MARC-145 cells are not infected by most NADC30-like and NADC34-like PRRSV strains. The essential scavenger receptor CD163 has been proved to mediate productive infection of PRRSV in various non-permissive cell lines. In this study, we systematically tested the porcine CD163 stably expressing 3D4/21 cells for infections with various PRRSV strains. The results showed that the porcine CD163-expressing macrophages support the infections of PRRSV2 of lineages 1, 5, and 8, as evidenced by Western blotting, immunofluorescence assay, quantitative PCR, and virus titration assay. Considering the current prevalence of NADC30-like and NADC34-like PRRSV2 of lineage 1 in China, the CD163-expressing macrophages are very useful for PRRSV research and disease management.

## 1. Introduction

Porcine reproductive and respiratory syndrome (PRRS) is characterized by reproductive failure in sows in the third trimester of pregnancy and respiratory symptoms in pigs of all ages, causing significant economic losses to the global swine industry [1]. PRRSV, as the causative agent, is an enveloped single-stranded positive-sense RNA virus and belongs to the genus *Betaarterivirus*, family *Arteriviridae*, order *Nidovirales* [2]. The genome of PRRSV is approximately 15.4 kb in length and encodes 16 nonstructural proteins (nsp1α, nsp1β, nsp2, nsp2TF, nsp2N, nsp3-6, nsp7α, nsp7β, and nsp8-12) and 8 structural proteins (GP2a/ORF2a, GP3/ORF3, GP4/ORF4, GP5/ORF5, E/ORF2b, 5a/ORF5a, M/ORF6, and N/ORF7) [3]. All PRRSV strains are classified into species PRRSV1 and PRRSV2 based on genomic variations [4]. In China, PRRSV2 strains are predominantly prevalent [5], and highly pathogenic variants of PRRSV2 (HP-PRRSVs) cause a serious loss to the pig industry throughout the country [6].

PRRSV infection shows a strongly restricted tropism for host species and target cells [7]. The porcine alveolar macrophages (PAMs) are the PRRSV primary target cells in vivo [7]. In addition, African green monkey kidney epithelial cell MA-104 and its derivatives, MARC-145 and CL2621 cells, support viral infection in vitro [8]. The specific cell tropism of PRRSV is mediated by the viral surface structural proteins and host cell receptor(s) [9]. To date, several independents but functionally related PRRSV receptors have been reported and the scavenger receptor CD163 is the key factor for PRRSV infection via promoting viral uncoating and internalization in macrophages [10]. 

CD163 is a 130 kDa type I transmembrane glycoprotein with a short cytoplasmic tail, a single transmembrane segment, and a large ectodomain consisting of nine scavenger receptor cysteine rich (SRCR) motifs [11]. Furthermore, the expression of CD163 in non-permissive cells such as SV40-transformed 3D4/21 [12], CHO and PK15 cells [13], BHK-21 cells [14], and murine macrophage-derived cells [15] has been shown to endow these cells permissiveness with PRRSV infection and replication. Recent studies have demonstrated that gene-edited pigs lacking functional CD163 are completely resistant to PRRSV, which confirms that CD163 is the primary receptor and plays an essential role in PRRSV infection [16,17,18]. 

In this study, a monoclonal porcine CD163 stable 3D4/21 cell line was systematically evaluated for the susceptibility to various PRRSV strains. The CD163 stably expressing cells were supportive to all tested PRRSV2 infections, and notably, the cells were sufficient to support NADC30-like and NADC34-like PRRSV proliferations, thus deepening the understanding CD163 function related to PRRSV infection.

## 2. Materials and Methods

### 2.1. Cells and Viruses

MARC-145 cells and 293T cells were cultured in DMEM (Life Technologies Corp., Grand Island, NY, USA) supplemented with 10% fetal bovine serum (FBS) and 100 U/mL of penicillin plus 100 μg/mL streptomycin. Porcine alveolar macrophages (3D4/21, ATCC CRL-2843) were cultured in RPMI (Hyclone Laboratories, Logan, UT, USA) containing 10% FBS with penicillin/streptomycin. All cells were maintained at 37 °C with 5% CO_2_ in a humidified incubator. PRRSV strains used in this study were all stored in our laboratories including the following: HP-PRRSV2 virulent strain XJ17-5 (lineage 8, GenBank: MK759853.1) and HP-PRRSV2 avirulent strain JSTZ1712-12 (lineage 8, GenBank: MK906026.1) [19,20]; Classic PRRSV2 CH-1a-like strain SD1612-1 (lineage 8, GenBank: MN119304.1) [21]; Classic PRRSV2 VR-2332-like strain R98 (lineage 5, GenBank: DQ355796.1); NADC30-like PRRSV2 strain SD17-38 (lineage 1, GenBank: MH068878.1) [22]; NADC34-like PRRSV2 strain Anheal-1 (lineage 1, GenBank: MH370474.1, a courtesy from Dr. Xizhao Chen at Beijing Anheal Laboratories Co. Ltd.); and PRRSV1 strain HLJB1 (GenBank: KT224385).

### 2.2. Establishment of Porcine Macrophages 3D4/21 Stably Expressing Porcine CD163

The porcine macrophages 3D4/21 cell line stably expressing porcine CD163 (pCD163) were constructed as we described previously [23]. Briefly, The CD163-EGFP-expressing lentiviruses were generated by co-transfecting pCD163-EGFP lentiviral vector with package plasmids psPAX2 and pMD2.G into 293T cells using Lipofectamine 2000 (Thermo Fisher Scientific, Shanghai, China). The supernatant containing pCD163-EGFP-expressing lentiviruses were used to infect the 3D4/21 cells, and the infected cells were selected with 2 µg/mL puromycin, by replacing with fresh medium every 3–4 days. About 2 weeks later, individual cell clones were screened for GFP/CD163 expression and supporting PRRSV replication. The obtained cell clone was further subjected two rounds of sub-cloning and screening by limited dilution method. 

### 2.3. Generation of Recombinant HP-PRRSV2 Avirulent Strain JSTZ1712-12 Carrying EGFP or DsRed

The recombinant PRRSV rJSTZ1712-12-EGFP/DsRed viruses were generated using the same methods as we generated recombinant HP-PRRSV XJ17-5-EGFP/dsRed previously [23,24]. Firstly, the full-genome cDNA clone of the PRRSV JSTZ1712-12 strain was constructed as we previously described [25]. Three overlapping fragments (F1, F2, and F3) spanning the full-length genome of JSTZ1712-12 were produced by PCR amplification using a set of primers (Appendix A). The full-length JSTZ1712-12 cDNA clone (rJSTZ1712-12) was obtained by sequential cloning of three segments using 4 unique restriction enzyme sites (*Pac*I, *Afl*II, *Asc*I and *Not*I) in pACYC177 (Appendix A). For generation of EGFP/dsRed PRRSV, the synthesized *Asc*I-*Kpn*I-EGFP-*Bcl*I-*Bgl*II fragment was first cloned into the *Asc*I-*Bgl*II sites of old pACYC177-JSTZ1712-12-F3 to obtain the modified F3 fragment. Next, the full length rJSTZ1712-12-GFP was generated by sequential cloning of F1, F2, and new F3 in pACYC177 (Appendix A). The full length rJSTZ1712-12-dsRed were generated by substitution of EGFP with dsRed within *Kpn*I-*Bcl*I sites of rJSTZ1712-12-GFP (Appendix A). The GFP/dsRed rPRRSV maintained a “green” or “red” phenotype throughout cell culture passages (Appendix A). 

### 2.4. Western Blot Analysis

Cells were collected and lysed in radio-immunoprecipitation assay (RIPA) buffer (50 mM Tris pH 7.2, 150 mM NaCl, 1% sodium deoxycholate, 1% Triton X-100). The protein samples were separated on 12% SDS-PAGE gels and transferred to polyvinylidene fluoride (PVDF) membranes (Merck Millipore, Billica, MA, USA). Membranes were blocked with 5% (*w*/*v*) nonfat milk in TBS containing 0.05% Tween (TBST) for 1 h at 37 °C. Rinsed blots were incubated with mouse anti-PRRSV N protein (N) antiserum, GFP mouse mAb (HT801-01, TransGen, Beijing, China), CD163 rabbit pAb (16646-1-AP, Proteintech, Wuhan, China), or GAPDH mouse mAb (AC002, ABclonal, Wuhan, China) (1:1000, each) at 37 °C for 1 h, and next with HRP-conjugated goat anti-mouse/rabbit IgG (1:10,000 in TBST, TransGen, Beijing, China) at 37 °C for 1h. Signals were detected by ECL chemiluminescent detection system (Tanon, Shanghai, China) according to the manufacturer’s instructions, and images were captured with Western blot imaging system (Tanon, Shanghai, China).

### 2.5. Immunofluorescence Assay (IFA)

MARC-145 cells and CD163-3D4/21 cells were seeded in 12-well plates and cells were either mock infected or infected with PRRSV at a multiplicity of infection of 0.1 for 48 h. The cells were fixed with 4% paraformaldehyde (Beyotime Biotech, Shanghai, China) for 15 min at room temperature, and then permeabilized with 0.1% Triton X-100 in PBS for 15 min, and blocked with 1% bovine serum albumin (BSA) in PBS for 1 h. After washing three times with PBS, cells were incubated with primary mouse nucleocapsid (N) mAb diluted in 1% BSA PBS at 37 ℃ for 3 h. The viral N mouse mAb 15A1 (a gift from Prof. Kegong Tian at National Research Center for Veterinary Medicine, China) was used for XJ17-5/JSTZ1712-12/SD16-1/R98 detection, whereas N mouse mAb 6A1 (Cat: JN0401, JNdiag, Beijing, China) was used for SD17-38/Anheal-1 detection. Cells were then washed and incubated with Goat anti-mouse IgG (H+L) secondary antibody, DyLight™ 594 (1:800; Cat: 35510, Thermo Fisher Scientific) for 1 h at room temperature in the dark. Cell nuclei were stained with 4′,6-diamidino-2-phenylindole (DAPI; Invitrogen, Waltham, MA, USA). All laser scanning images were obtained using a fluorescence microscope (Leica, SPE, Buffalo Grove, IL, USA).

### 2.6. Virus Titration

MARC-145 cells grown in 96-well plates were infected with 10-fold serial dilutions of PRRSV samples. After 1 h incubation at 37 °C, the supernatants were replaced with fresh DMEM containing 2% FBS. Five days post infection, the cytopathic effect (CPE) characterized by clumping and shrinkage of cells was obviously visible in MARC-145 cells and the viral titers, expressed as 50% tissue culture infective dose (TCID50), were calculated according to the method of Reed–Muench.

For lineage 1 virus titration, CD163-3D4/21 cells grown in 96-well plates were infected with 10-fold serial dilutions of PRRSV samples. After 72 h, cells were fixed and virus-infected cells were stained with PRRSV-specific N mAb (6A1) and Goat anti-mouse IgG (H+L) secondary antibody, DyLight™ 594. Based on fluorescence signals, virus titers were calculated by the method of Reed–Muench.

### 2.7. Quantitative Reverse Transcription Polymerase Chain Reaction (RT-qPCR) 

Total RNA was isolated from MARC-145 cells and CD163-3D4/21 cells using TRIzol reagent (Thermo Fisher Scientific, Shanghai, China) and then reverse transcribed into complementary DNA with HiScript^®^ 1st Strand cDNA Synthesis Kit (Vazyme, Nanjing, China) according to the manufacturer’s instructions. SYBR Green real-time PCR was performed to evaluate PRRSV RNA level. The amplification conditions were 95 °C for 30 s, followed by 40 cycles of 95 °C for 10 s and 60 °C for 30 s using StepOne Plus real-time PCR system (Applied Biosystems, Foster City, CA, USA). For the quantification of NADC30-like and NADC34-like PRRSV RNA levels, The TaqMan real-time PCR using Premix Ex Taq (Probe qPCR, 2×) (TaKaRa, Beijing, China) and the StepOne Plus Real-Time PCR System was performed. The amplification conditions were 95 °C for 30 s, followed by 40 cycles of 95 °C for 5 s and 60 °C for 1 min. The viral gene copy numbers were calculated according to the following formula: lg (copies/mL) = lg [2^(control group CT − experimental group CT)^]. Primers and TaqMan probe used for RT-qPCR are listed in Appendix A.

### 2.8. Statistical Analysis

The data are presented as the means ± standard deviations (SD, *n* = 3). Statistical significance between groups was determined by performing a Student’s *t* test with GraphPad Prism 6.0 software. The *p* value of <0.05 was considered statistically significant.

## 3. Results

### 3.1. CD163-3D4/21 Cells Are Susceptible to HP-PRRSV2 Virulent Strain XJ17-5 (Lineage 8) Infections

CD163-3D4/21 cells were first infected with highly pathogenic (HP) PRRSV virulent strain XJ17-5 and rescued XJ17-5 (rXJ17-5) at an MOI of 0.1. As shown in Figure 1A, upon two PRRSV infections, CD163-3D4/21 cells, but not non-permissive 3D4/21 cells, expressed viral N proteins detected by Western blotting (top and bottom, Figure 1A). In infected control MARC-145 cells, the viral N proteins were also easily detected by Western blotting (middle, Figure 1A). In fact, a typical cytopathic effect (CPE) was observed in PRRSV-infected CD163-3D4/21 cells as well as in MARC-145 cells (first columns, Figure 1B). The specificity of CPE was further confirmed by N specific fluorescent signals observed in CD163-3D4/21 cells upon both XJ17-5 and rXJ17-5 PRRSV2 infections, as in MARC-145 cells, indicating the spread of the virions to the neighboring cells and productive infections (third columns, Figure 1B). The viral genome and infectivity of two PRRSV2 were quantified by RT-qPCR detection of ORF7 transcription (Figure 1C) and TCID50 assay (Figure 1D), respectively. Both PRRSV2 possessed similar ORF7 transcription levels between CD163-3D4/21 and MARC-145 cells (Figure 1C). Consistently, both PRRSV2 achieved similar infection levels in CD163-3D4/21 and MARC-145 cells, but no detectable infection levels in non-permissive 3D4/21 cells (Figure 1D).

We next examined the infections of two rescued PRRSV2 carrying fluorescent proteins (rXJ17-5-EGFP and rXJ17-5-dsRed) in CD163-3D4/21 cells. In Western blotting (Figure 1A), the viral N proteins were barely detected in CD163-3D4/21, whereas two viral N proteins were obvious in infected MARC-145 cells (top and middle, Figure 1A). In RT-qPCR (Figure 1E) and TCID50 assay (Figure 1F), two fluorescent rPRRSV2 exhibited substantially lower levels of ORF7 transcriptions and infectivity in CD163-3D4/21 cells versus MARC-145 cells (Figure 1E,F). In non-permissive 3D4/21 cells, two fluorescent rPRRSV2 showed neither N protein expression (Bottom, Figure 1A) nor virus infectivity (Figure 1F). Altogether, the results demonstrated that porcine CD163 confers the non-permissive 3D4/21 cells susceptibility to different PRRSV2 XJ17-5 strains, with rPRRSV2 XJ17-5-EGFP/dsRed much less efficient.

### 3.2. CD163-3D4/21 Cells Are Susceptible to HP-PRRSV2 Avirulent Strain JSTZ1712-12 (Lineage 8) Infections

With the same purpose, CD163-3D4/21 cells were infected with avirulent JSTZ1712-12 and rescued JSTZ1712-12 at an MOI of 0.1. Upon two PRRSV2 infections, CD163-3D4/21, but not non-permissive 3D4/21 cells, expressed viral N proteins detected by Western blotting (top and bottom, Figure 2A), and in control MARC-145 cells, the viral N proteins were easily detected by Western blotting (middle, Figure 2A). Two JSTZ1712-12 PRRSV2 exhibited CPE in both CD163-3D21/4 cells and MARC-145 cells, and the immunofluorescence further confirmed the specificity of CPE by showing N-specific signals in the cell clusters (Figure 2B). Both PRRSV2 possessed close ORF7 transcription levels between CD163-3D4/21 and MARC-145 cells in RT-qPCR assay (Figure 2C). Consistently, both PRRSV2 achieved close infection levels between CD163-3D4/21 and MARC-145 cells, but no detectable infection level in non-permissive 3D4/21 cells (Figure 2D).

We also compared the infections of two rescued JSTZ1712-12 PRRSV2 carrying fluorescent proteins (rJSTZ1712-12-EGFP and rJSTZ1712-12-dsRed) in CD163-3D4/21 cells. In Western blotting, the viral N proteins were barely or much less detected in CD163-3D4/21, whereas in MARC-145 cells, the N proteins were equally expressed between rJSTZ1712-12-GFP/dsRed and JSTZ1712-12 (Figure 2A). In RT-qPCR and TCID50 assays, two fluorescent rPRRSV2 exhibited much lower levels of ORF7 transcriptions and infectivity in CD163-3D4/21 vs. MARC-145 cells (Figure 2E,F). In non-permissive 3D4/21 cells, two fluorescent rPRRSV2 showed neither N protein expression (bottom, Figure 2A) nor virus infectivity (Figure 2F). Together and again, the results demonstrated that porcine CD163 confers the non-permissive 3D4/21 cells susceptibility to avirulent PRRSV2 JSTZ1712-12 strains, with rPRRSV2 JSTZ1712-12-EGFP/dsRed less efficient.

### 3.3. CD163-3D4/21 Cells Are Able to Be Infected by Different Classical PRRSV2 Strains

We wondered what the susceptibility of CD163-3D4/21 cells to classical PRRSV is. CD163-3D4/21 cells were infected with PRRSV CH-1a-like strain SD1612-1 of lineage of 8 at an MOI of 0.1. As shown in Figure 3A, SD1612-1 expressed N proteins in both CD163-3D4/21 and MARC-145 cells. Further, the typical CPE and N specific fluorescence signals were observed in SD1612-1 infected CD163-3D4/21 cell as well as MARC-145 cells (Figure 3B). However, RT-qPCR showed that the transcription level of viral ORF7 in CD163-3D4/21 cells was significantly lower than that in MARC-145 cells (Figure 3C). The virus titer of PRRSV in CD163-3D4/21 cells was much lower than that in MARC-145 cells (Figure 3D).

The BJ-4-like/VR2332-like classical PRRSV2 of lineage 5 appeared as early as 1996, but it has been always non-pandemic in China with low clinical detection rate [5]. CD163-3D4/21 cells were tested for infection with PRRSV R98 strain of lineage 5 at an MOI of 0.1. R98 strain expressed N proteins in both CD163-3D4/21 and MARC-145 cells (Figure 4A). The visible CPE and viral N specific fluorescent signals appeared in R98 infected CD163-3D4/21 and infected MARC-145 cells (Figure 4B). Additionally, R98 exhibited similar viral ORF7 transcriptions by RT-qPCR (Figure 4C) and similar viral infectivity by TCID50 assay (Figure 4D) in both CD163-3D4/21 and MARC-145 cells (Figure 4C,D). Taken together, the results demonstrated that porcine CD163 expressing macrophages are susceptible to different classical PRRSV2 strains.

### 3.4. CD163-3D4/21 Cells Are Able to Be Infected by NADC30-like and NADC34-like PRRSV2 Strains

In 2012, the existence of NADC30-like strains was reported for the first time in pig herds in China [26]. These strains were prevalent in pig herds around 2015 and have become the main epidemic strains currently. Moreover, the NADC34-like PRRSV2 occurred in at least four different regions of China [27,28], suggesting that NADC34-like PRRSV2 has been becoming endemic and epidemic in China [28]. Therefore, the CD163-3D4/21 cells were tested for infections by NADC30-like PRRSV2 strain SD17-38 and NADC34-like PRRSV2 strain Anheal-1. First, as shown in Figure 5A, both SD17-38 and Anheal-1 expressed viral N proteins in infected CD163-3D4/21 cells but not in MARC-145 cells. Second, in infected CD163-3D4/21 cells, but not in MARC-145 cells, the CPE and viral N fluorescent signals were observed with both SD17-38 and Anheal-1 infections (Figure 5B). Third, The ORF6 transcription of NADC30-like strain SD17-38 reached plateau at 24 h post infection, whereas ORF6 transcription of NADC34-like strain Anheal-1 gradually peaked at 72 h post infection (Figure 5C). Fourth, the viral titers of SD17-38 and Anheal-1 were measured in CD163-3D4/21 cells based on viral N fluorescence. The results showed that SD17-38 reached a peak TCID50 level of 10^6.5^ at 48 h post infection, whereas Anheal-1 kept increasing up to 72 h post infection but with 1–3 log lower in TCID50 titers (Figure 5D). Taken together, the data demonstrated that CD163-3D4/21 cells are also susceptible to PRRSV2 of lineage 1 including NADC30-like and NADC34-like strains.

## 4. Discussion

PRRSV has caused devastating economic losses in the pig industry worldwide; however, the understanding of PRRSV pathogenesis and vaccine development have been limited by the lack of effective cell lines and small animal models [1]. Since PRRSV exhibits a strict preference for cells of the monocyte–macrophage lineage, the differentiated primary porcine alveolar macrophages (PAMs) are the main target cells for virus replication in vivo [29]. However, due to batch variation, risk of pathogen contamination, and high economic cost, this type of cells cannot be used for live vaccine production. In addition to primary PAMs, immortalized monkey kidney cell lines such as MARC-145 have been shown to be permissive for PRRSV infection, and were a key platform for PRRSV virus isolation and vaccine development [13]. The MARC-145 cell line has overcome many of the problems associated with primary PAMs and are routinely used for large-scale production of PRRSV [13]. However, PRRSV has evolved with different lineages and many PRRSV strains failed to infect MARC-145 cells [30,31,32]. Additionally, because of the monkey origin, PRRSV proliferated in MARC-145 cells easily result in genetic alterations [33], and thus are not ideal for PRRSV delicate function study.

So far, several receptors involved in PRRSV infection have been described, including heparin sulfate, sialoadhesin (CD169), CD163, CD151, vimentin, DC-SIGN (dendritic cell-specific intercellular adhesion molecule-3-grabbing non-integrin/CD209), and MYH9 (non-muscle myosin heavy chain 9) [34]. Heparin sulfate (HS) serves as an attachment factor for PRRSV [35]. Sialoadhesin (Sn) has been identified to interact with PRRSV surface M/GP5 complex and mediate viral internalization [36]. Most importantly, scavenger receptor CD163 is the key factor for PRRSV infection via binding with PRRSV surface GP2a/3/4 complex and promoting viral internalization and uncoating in macrophages [10]. CD163 alone was sufficient for conferring permissiveness to multiple PRRSV2 non-permissive cells (including porcine kidney, feline kidney and baby hamster kidney) [14]. Furthermore, porcine kidney cells expressing CD163 could support replication of PRRSV2 strain BJ-4 or HN07-1 [37]. PAM cell lines stably expressing CD163 were fully permissive for both PRRSV1 Lelystad strain and PRRSV2 VR-2332 strain [12]. However, the infections of porcine macrophage cell lines expressing CD163 with different lineages of PRRSV have not been systematically investigated. Here, we infected porcine macrophage 3D4/21 expressing CD163 with PRRSV2 of different lineages (lineages 1, 5, 8) and PRRSV1. Our data showed that except PRRSV1 strain HLJB1, CD163-3D4/21 cells are able to support all PRRSV2 of lineages 1, 5, and 8. The 3D4/21 cell line was originally from immortalized primary porcine alveolar macrophage (PAMs) and not susceptible to PRRSV anymore as primary PAMs [38]. During the immortalization by SV40 LT, the CD163 has disappeared and after reintroduction of porcine CD163, the stable 3D4/21 cells become susceptible to PRRSV again (Appendix A).

In China, CH-1a-like classical PRRSV2 (lineage 8) was first isolated in 1995 [39]. In 2006, the outbreak of JXA1-like highly pathogenic PRRSV2 (HP-PRRSV2, lineage 8) caused enormous economic losses in China [40]. In 2010, a QYYZ-like PRRSV2 (lineage 3) isolate was reported [41]. Since 2013, NADC30-like PRRSV2 (lineage 1) variants have been frequently encountered in China [42]. The genome of NADC30-like PRRSV2 is highly variable and had low homology with classical (CA)-PRRSV and highly pathogenic (HP)-PRRSV. Further, it is easy to recombine with other PRRSV to form new strains, endowed with more complex genome characteristics, cell tropism, and pathogenicity [26]. Notably, studies in recent years have shown that NADC30-like PRRSV from clinical samples has gradually increased, reaching a similar or even higher isolation rate relative to HP-PRRSV and becoming the prevalent lineage of PRRSV strain in China [5,26]. In 2017, NADC34-like PRRSV2 (lineage 1) was reported for the first time in China, and since then, it has been isolated and detected in different regions in China [28]. Here, we observed that CD163-3D4/21 cells were able to support the infections of all tested PRRSV2 strains, including NADC30-like PRRSV (SD17-38) and NADC34-like PRRSV (Anheal-1), in spite of their failure to infect MARC-145 cells, which is the first report as far as we know.

Additionally, we noticed the large variations of different PRRSV2 strains during infections of CD163-3D4/21 cells versus MARC-145 cells. For example, NADC30-like SD17-38 and NADC34-like Anheal-1 replicated in CD163-3D4/21 cells but not in MARC-145 cells; HP virulent XJ17-5/rXJ17-5, HP avirulent JSTZ1712-12/rJSTZ1712-12 and classical R98 replicated similarly in both cell types; HP virulent rXJ17-5-EGFP/dsRed, HP avirulent rJSTZ1712-12-EGFP/dsRed and classical SD1612-1 replicated in CD163-3D4/21 cells with much less efficiency relative to MARC-145 cells. Currently, we do not know the exact reason for these large variations; however, we think the underlying reason may come from two aspects. One is the high heterogeneity existing in different PRRSV strains, and another is the different cellular environment existed between CD163-3D/21 and MARC-145 cells. The complex interactions of viral and cellular factors resulted in the large variations in viral infections.

In summary, we systematically evaluated monoclonal porcine macrophages 3D4/21 stably expressing CD163 for infections with PRRSV of different lineages. The stable macrophages are susceptible to different PRRSV2 infections, including NADC30-like and NADC34-like PRRSV strains, which do not replicate in MARC-145 cells. Therefore, the macrophage cell line may substitute for or complement with currently widely used MARC-145 cells and will be a valuable tool to study the PRRSV biology and/or produce large scale PRRSV.

## Figures and Tables

**Figure 1 viruses-14-02056-f001:**
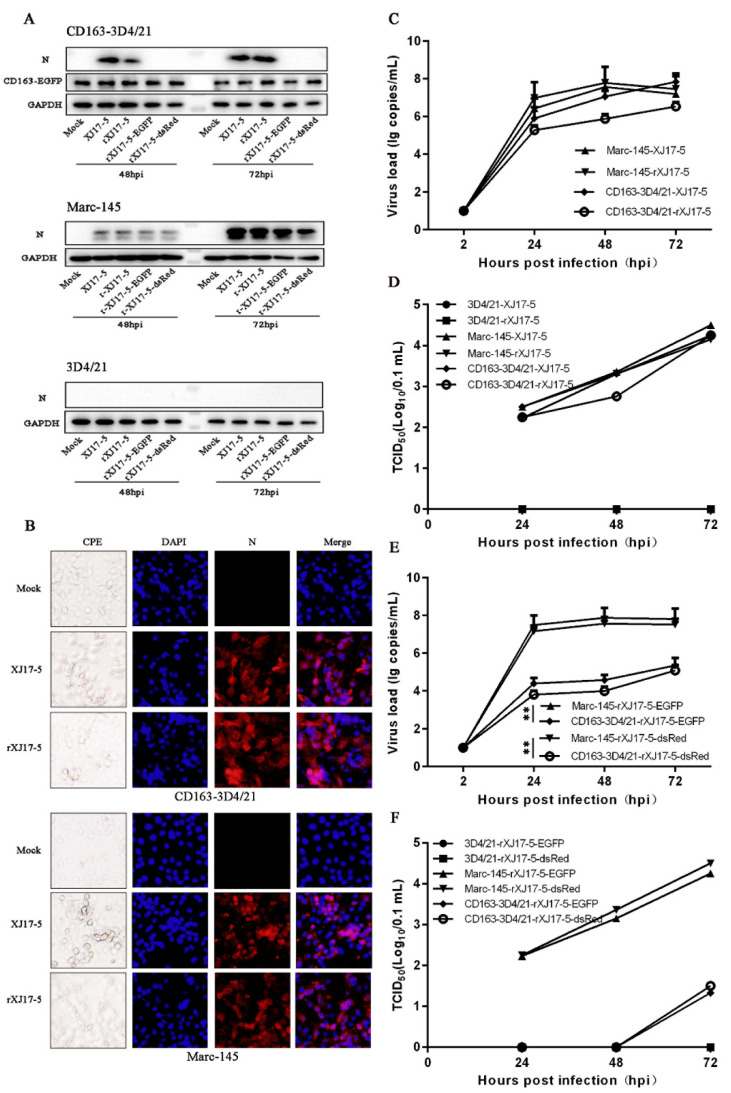
Growth of HP-PRRSV2 virulent strain XJ17-5 (lineage 8) on 3D4/21 cells stably expressing porcine CD163. (**A**) The CD163-3D4/21 cells, MARC-145 cells, and 3D4/21 cells seeded in 24-well plates were infected with PRRSV XJ17-5 and infectious cDNA clones (rXJ17-5, rXJ17-5-EGFP, or rXJ17-5-dsRed) at an MOI of 0.1, for 48, and 72 h, respectively, followed by Western blotting. (**B**) CD163-3D4/21 cells and MARC-145 cells were infected with PRRSV XJ17-5 and rXJ17-5 at an MOI of 0.1, respectively. The infected cells were observed for CPE and fixed at 48 h post infection (hpi) followed by immunofluorescence assay (IFA): PRRSV N protein specific mouse mAb (15A1, 1:500) was used as the primary antibody, and the DyLight™ 594 (Goat anti-mouse IgG, 1:800, Invitrogen) was used as the secondary antibody. (**C**–**F**) Replication of PRRSV XJ17-5 and infectious cDNA clones (rXJ17-5, rXJ17-5-EGFP, or rXJ17-5-dsRed) in CD163-3D4/21 cells, MARC-145 cells, or 3D4/21 cells. The ORF7 transcriptions and virus titers of PRRSV XJ17-5 and infectious cDNA clones of XJ17-5 within 72 hpi were determined by SYBR Green real-time RT-PCR assay (**C**,**E**) and TCID50 (**D**,**F**), respectively. ** *p* < 0.01.

**Figure 2 viruses-14-02056-f002:**
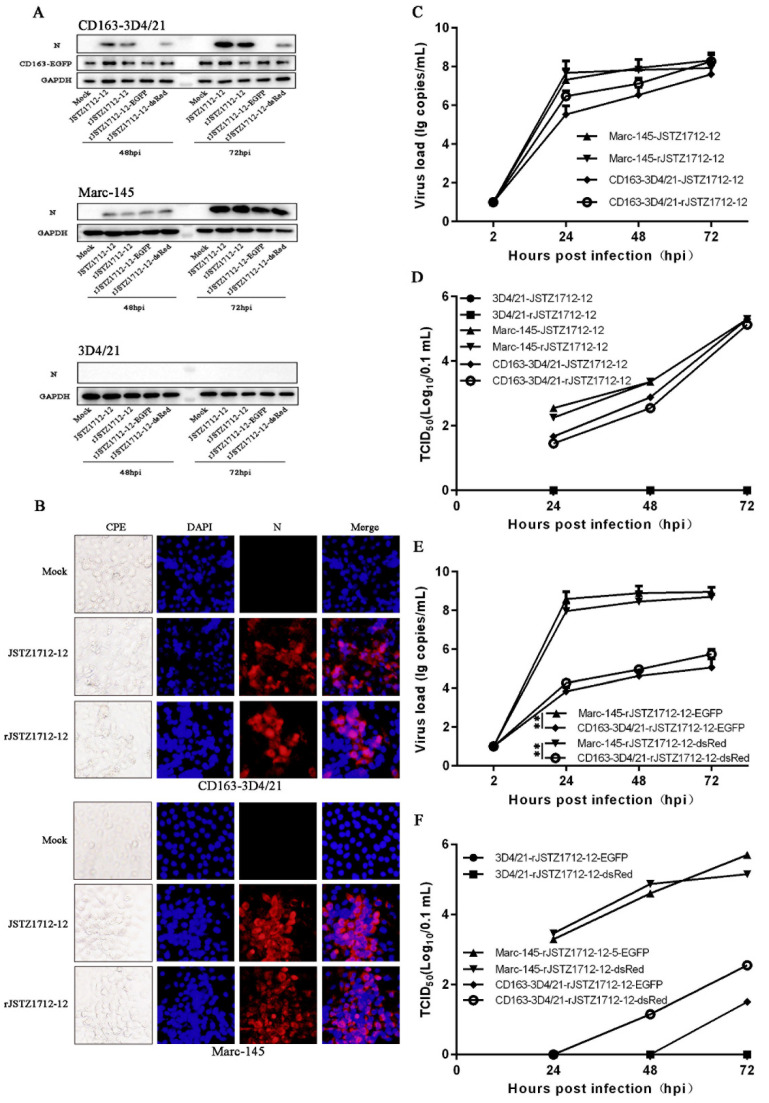
Growth of HP-PRRSV2 avirulent strain JSTZ1712-12 (lineage 8) on 3D4/21 cells stably expressing porcine CD163. (**A**) The CD163-3D4/21 cells, MARC-145 cells, and 3D4/21 cells seeded in 24-well plates were infected with PRRSV JSTZ1712-12 and infectious cDNA clones (rJSTZ1712-12, rJSTZ1712-12-EGFP, or rJSTZ1712-12-dsRed) at an MOI of 0.1, for 48 and 72 h, respectively, followed by Western blotting. (**B**) CD163-3D4/21 cells and MARC-145 cells were infected with PRRSV JSTZ1712-12 and rJSTZ1712-12 at an MOI of 0.1, respectively. The infected cells were observed for CPE and fixed at 48 hpi for IFA using PRRSV N mouse mAb (15A1). (**C**–**F**) Replication of PRRSV JSTZ1712-12 and infectious cDNA clones (rJSTZ1712-12, rJSTZ1712-12-EGFP, or rJSTZ1712-12-dsRed) in CD163-3D4/21 cells, MARC-145 cells, or 3D4/21cells. The ORF7 transcriptions and virus titers of PRRSV JSTZ1712-12 and infectious cDNA clones within 72 hpi were determined by SYBR Green real-time RT-PCR (**C**,**E**) and TCID50 assay (**D**,**F**), respectively. ** *p* < 0.01.

**Figure 3 viruses-14-02056-f003:**
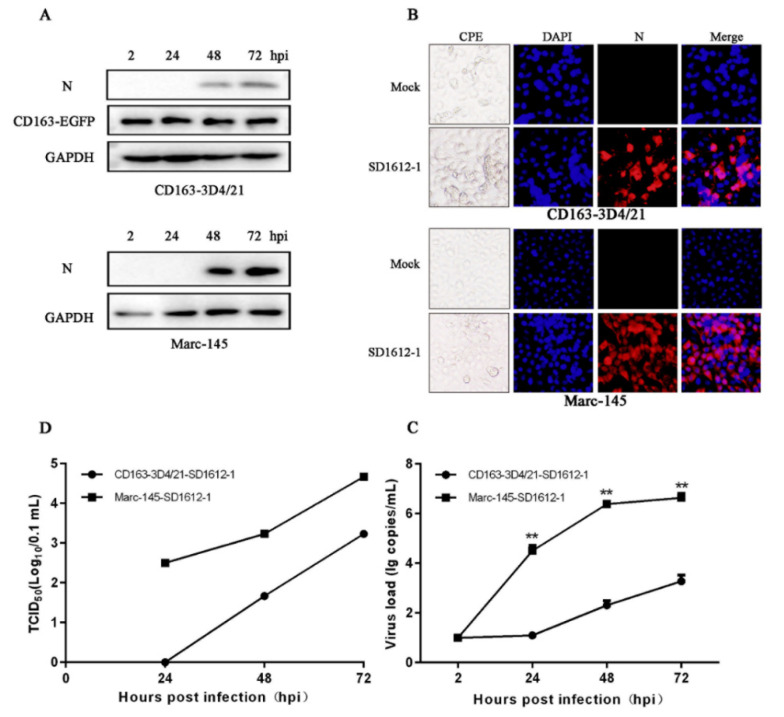
Growth of classical PRRSV2 strain SD1612-1 (lineage 8) on 3D4/21 cells stably expressing porcine CD163. (**A**) CD163-3D4/21 cells and MARC-145 cells seeded in 24-well plates were infected with PRRSV SD1612-1 at an MOI of 0.1, for the indicated time points, followed by Western blotting. (**B**) The infected cells were observed for PRRSV specific CPEs and fixed at 48 hpi followed by IFA using N mouse mAb (15A1). (**C**,**D**) The ORF7 transcriptions and viral titers of PRRSV SD1612-1 in CD163-3D4/21 cells and MARC-145 cells within 72 hpi were determined by SYBR Green real-time RT-PCR (**C**) and TCID50 assay (**D**), respectively. ** *p* < 0.01.

**Figure 4 viruses-14-02056-f004:**
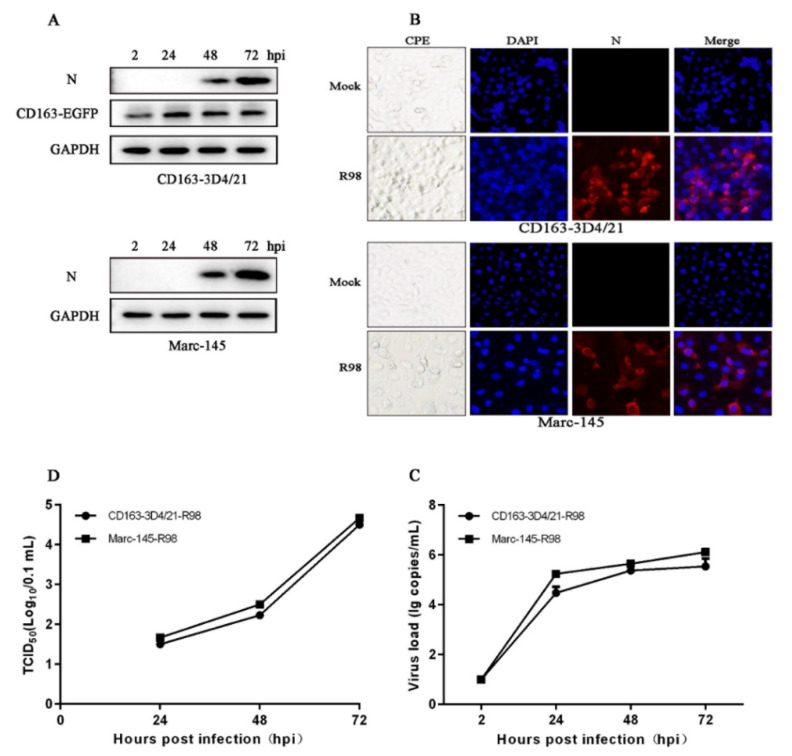
Growth of classical PRRSV2 strain R98 (lineage 5) on 3D4/21 cells stably expressing porcine CD163. (**A**) The CD163-3D4/21 cells and MARC-145 cells seeded in 24-well plates were infected with PRRSV R98 at an MOI of 0.1, for the indicated time points, followed by Western blotting. (**B**) The infected cells were observed for PRRSV specific CPE and fixed at 48 hpi followed by IFA using PRRSV N specific mAb (15A1). (**C**,**D**) The ORF7 transcriptions and viral titers of PRRSV R98 in CD163-3D4/21 cells and MARC-145 cells within 72 hpi were determined by SYBR Green real-time RT-PCR (**C**) and TCID50 assay (**D**), respectively.

**Figure 5 viruses-14-02056-f005:**
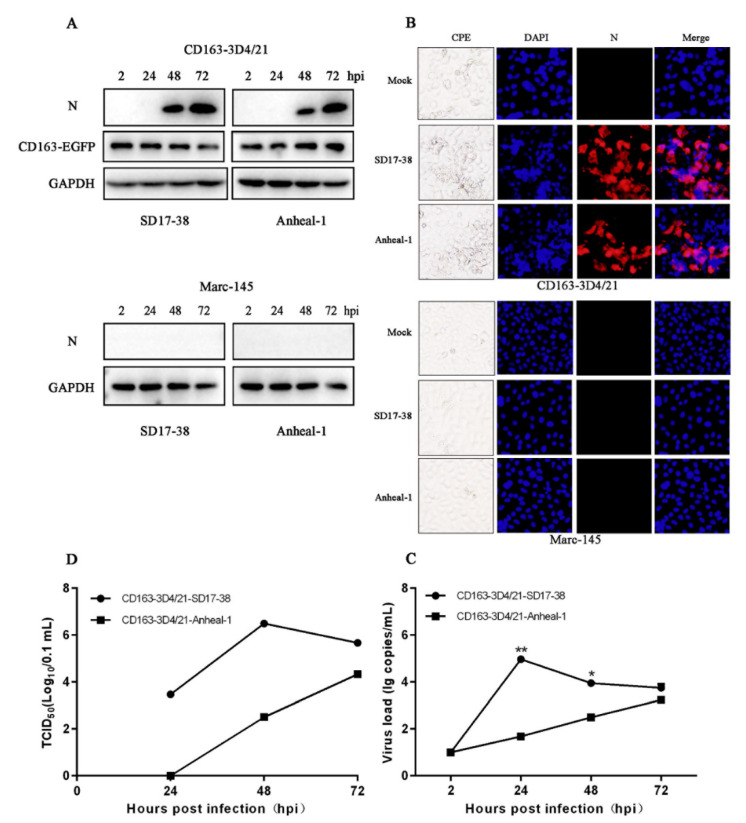
CD163-3D4/21 cells are able to be infected by NADC30-like and NADC34-like PRRSV2 strains. (**A**) The CD163-3D4/21 cells and MARC-145 cells seeded in 24-well plates were infected with NADC30-like SD17-38 and NADC34-like Anheal-1, at an MOI of 0.1, for the indicated time points, followed by Western blotting. (**B**) The infected cells were observed for CPE, and fixed at 48 hpi followed by IFA using PRRSV N specific mouse mAb (6A1, 1:500 dilution). (**C**) The ORF6 gene transcriptions of NADC30-like SD17-38 and NADC34-like Anheal-1 were determined by TaqMan real time RT-PCR. (**D**) The viral titers of NADC30-like SD17-38 and NADC34-like Anheal-1 in CD163-3D4/21 cells were determined by TCID50 assay based on viral N fluorescence signals. * *p* < 0.05, ** *p* < 0.01.

## Data Availability

The raw data presented in this study are available online.

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
