# Peer review of "CD163-Expressing Porcine Macrophages Support NADC30-like and NADC34-like PRRSV Infections"

_viruses, 2022, doi:10.3390/v14092056_

Round 1

Reviewer 1 Report

1. The manuscript by Yulin and colleagues describes the evaluation of a porcine macrophage cell line 3D4/21 which stably expresses pCD163 for the susceptibility of infections by various PRRSV strains. Their results showed that the cell line supports the propagation of PRRSV2 lineages 1, 5, 8; and especially the NADC30-like PRRSV2 strains in lineage 1, which usually do not grow well in Marc-145 cells. The data is solid and well presented, and the results provide novel information for PRRSV research and vaccine development. 

2. The abstract: need to give a brief summary of the results of the WB, IFA, titration and qPCR.

3. Page 13, 2nd paragraph, line 3, change the “conducive” to “susceptible”.

Author Response

1. The manuscript by Yulin and colleagues describes the evaluation of a porcine macrophage cell line 3D4/21 which stably expresses pCD163 for the susceptibility of infections by various PRRSV strains. Their results showed that the cell line supports the propagation of PRRSV2 lineages 1, 5, 8; and especially the NADC30-like PRRSV2 strains in lineage 1, which usually do not grow well in Marc-145 cells. The data is solid and well presented, and the results provide novel information for PRRSV research and vaccine development.

Thanks very much for your appreciation.

2. The abstract: need to give a brief summary of the results of the WB, IFA, titration and qPCR.

Answer: The information of WB, IFA, titration and qPCR has been added in the Abstract. 

3. Page 13, 2nd paragraph, line 3, change the “conducive” to “susceptible”.

Answer: it has been changed. 

Reviewer 2 Report

The authors established a porcine macrophage cell line 3D4/21 stably expressing scavenger receptor CD163, a receptor/uncoating factor for PRRSV, and investigated if the cells can support replication of PRRSV of different lineages, including those in the current circulation in China. Previously, it was shown that expression of CD163 confers permissiveness for PRRSV infection to different cell lines. In the current study, all tested PRRSV strains could replicate in the new cell line, including SD17-38 and Anheal-1 which do not replicate in MARC-145 cells, a standard system for PRRSV propagation. The experiments are performed on a high technical level and are clearly presented. There are a couple of issues the authors need to address to improve the manuscript:

Fig. 1 and 2. The authors should at least speculate in the Discussion why the PRRSV derivatives expressing fluorescent proteins are severely compromised in CD163-3D4/21 cell line while the parental viruses replicate robustly, and both the parental viruses and the fluorescent proteins expressing derivatives are fully replication competent in MARC-145. This is one of the most interesting results of this study which may lead to important discoveries about the molecular mechanisms of PRRSV replication

Fig. 5C. What is the rationale for comparing transcription of different genes of different viruses (ORF6 for SD17-38 strain and ORF3 for Aheal-1)?  The authors should either provide a convincing argument why such analysis is relevant, or perform quantitation of expression of the same genes for both viruses.

Author Response

The authors established a porcine macrophage cell line 3D4/21 stably expressing scavenger receptor CD163, a receptor/uncoating factor for PRRSV, and investigated if the cells can support replication of PRRSV of different lineages, including those in the current circulation in China. Previously, it was shown that expression of CD163 confers permissiveness for PRRSV infection to different cell lines. In the current study, all tested PRRSV strains could replicate in the new cell line, including SD17-38 and Anheal-1 which do not replicate in MARC-145 cells, a standard system for PRRSV propagation. The experiments are performed on a high technical level and are clearly presented. There are a couple of issues the authors need to address to improve the manuscript:

Thanks for the appreciation and suggestions.

Fig. 1 and 2. The authors should at least speculate in the Discussion why the PRRSV derivatives expressing fluorescent proteins are severely compromised in CD163-3D4/21 cell line while the parental viruses replicate robustly, and both the parental viruses and the fluorescent proteins expressing derivatives are fully replication competent in MARC-145. This is one of the most interesting results of this study which may lead to important discoveries about the molecular mechanisms of PRRSV replication

Answer: Thanks. We do not know the exact reason now. However, we think it is from both the viral and cellular factors which have complex interaction, giving the large variations in virus infections. We have added one paragraph in Discussion to talk about the details.

Fig. 5C. What is the rationale for comparing transcription of different genes of different viruses (ORF6 for SD17-38 strain and ORF3 for Aheal-1)?  The authors should either provide a convincing argument why such analysis is relevant, or perform quantitation of expression of the same genes for both viruses.

Answer: Thanks for comment. The old ORF6 and ORF3 primers/probes were specifically used for SD17-38 and Aheal-1, respectively, which may not be ideal for the comparison in this case. Therefore, we chose a set of new ORF6 primers/probe which are common for both SD17-38 and Aheal-1, and the Fig 5C was updated with new experimental results.
